# Molecular Identification of Fungal Species through Multiplex-qPCR to Determine Candidal Vulvovaginitis and Antifungal Susceptibility

**DOI:** 10.3390/jof9121145

**Published:** 2023-11-27

**Authors:** Inés Arrieta-Aguirre, Pilar Menéndez-Manjón, Giulia Carrano, Ander Diez, Íñigo Fernandez-de-Larrinoa, María-Dolores Moragues

**Affiliations:** 1Department of Nursing I, Faculty of Medicine and Nursing, University of the Basque Country UPV/EHU, 48940 Leioa, Biscay, Spain; pilarmmt@gmail.com (P.M.-M.); lola.moragues@ehu.es (M.-D.M.); 2Department of Immunology, Microbiology and Parasitology, University of the Basque Country UPV/EHU, 48940 Leioa, Biscay, Spain; giulia.carrano@ehu.eus (G.C.); ander.diez@ehu.eus (A.D.); 3Applied Chemistry, University of the Basque Country UPV/EHU, 20018 Donostia-San Sebastian, Gipuzkoa, Spain; if.larrinoa@ehu.eus; 4IIS BioCruces Bizkaia, 48903 Barakaldo, Biscay, Spain

**Keywords:** vulvovaginal candidiasis (VVC), qPCR, *Candida* species, antifungal susceptibility, resistance-related mutations

## Abstract

Vulvovaginal candidiasis (VVC) is a prevalent condition affecting women worldwide. This study aimed to develop a rapid qPCR assay for the accurate identification of VVC etiological agents and reduced azole susceptibility. One hundred and twenty nine vaginal samples from an outpatient clinic (Bilbao, Spain) were analyzed using culture-based methods and a multiplex qPCR targeting fungal species, which identified *Candida albicans* as the predominant species (94.2%). Antifungal susceptibility tests revealed reduced azole susceptibility in three (3.48%) isolates. Molecular analysis identified several mutations in genes associated with azole resistance as well as novel mutations in *TAC1* and *MRR1* genes. In conclusion, we developed a rapid multiplex qPCR assay that detects *C. albicans* in vulvovaginal specimens and reported new mutations in resistance-related genes that could contribute to azole resistance.

## 1. Introduction

Vulvovaginal candidiasis (VVC) is a common condition characterized by vulvovaginal inflammation in combination with the detection of one or more *Candida* or fungal species [1]. Approximately 75% of women will experience at least one episode of VVC in their lifetime [2], while some of them will develop recurrent VVC (RVVC), which is arbitrarily defined as four or more episodes every year; therefore, VVC is a non-invasive fungal disease affecting millions of women worldwide [3,4]. Even though it is not a life-threatening infection, for some women, VVC implies a severe loss of life quality, mainly due to the wide variety of clinical symptoms, which include vulvovaginal pruritus, soreness, or dyspareunia, and also due to the fact that acute recurrences often lead to depression and anxiety [5]. Although it is widely recognized that the misdiagnosis of patients with vaginal complaints is extremely common, the lack of an accurate diagnosis is still widespread [6]. Most VVC cases are caused by *Candida albicans*; nevertheless, a mycological shift in etiologic species has been reported; *Candida glabrata*, *Candida parapsilosis*, *Candida tropicalis*, and *Candida krusei* are the main species among the non-*albicans Candida* (NAC) [7]. In this regard, it is essential to bear in mind that apart from clinical manifestations, the diagnosis of VVC should involve laboratory methods such as microscopy or a culture test; the latter is especially necessary in case of RVVC or infections caused by NAC species [8]. Culture is the gold standard for *Candida* vulvovaginitis diagnosis, but it delays the results for 48–96 h [6]. This fact and the availability of topical over-the-counter (OTC) drugs compel patients to self-diagnose and self-medicate, ultimately leading to a bias in the epidemiological data and an increase in less-drug-susceptible isolates [4]. The latter may occur due to the appearance of NAC isolates, as some of them are less susceptible or intrinsically resistant to azoles, particularly *C. glabrata*, *C. tropicalis*, or *C. krusei* [3]. In this context, some authors have described a trend shift towards NAC species in vaginal samples [9]. Another possibility is the increase in *C. albicans*-resistant isolates, and recent data show a worryingly high level of resistance to various azoles in vulvovaginal isolates [9,10]. Azole antifungals remain the treatment of choice for uncomplicated VVC: topical fluconazole, clotrimazole, or miconazole vaginal creams, or oral fluconazole [11]. Sustaining the use of fluconazole over prolonged periods stands as the first line of treatment for RVVC and has been demonstrated to enhance the quality of life of 96% of treated women [12]. Nevertheless, this long-term treatment with fungistatic fluconazole may entail ideal conditions for *C. albicans* strains to evolve into a resistant phenotype [4]. The main trajectory for resistance development is the upregulation of *CDR1*, *CDR2*, or *MDR1* genes related to efflux pumps. In addition, the overexpression of or mutations in the *ERG11* gene (encoding 14-α demethylase) and loss-of-function mutations in the *ERG3* gene (encoding Δ5,6 desaturase), both involved in ergosterol synthesis, are frequently reported in *C. albicans*-resistant isolates [9,13]. Furthermore, gain-of-function (GOF) mutations in *TAC1*, *UPC2*, *MRR1* [14,15,16], and, more recently, *MRR2* genes [17], transcription regulators of some of the genes mentioned above lead to constitutive overexpression, resulting in increased resistance. Additionally, genome aneuploidies and loss of heterozygosity are also involved in azole resistance [18]. Many *C. albicans* isolates exhibit a combination of resistance mechanisms resulting in high levels of drug resistance and clinical failure [19].

Based on isolates identified in VVC, here, we report a real-time PCR assay targeting the multi-copy ITS1 gene of six yeasts species, namely, *C. albicans*, *C. parapsilosis*, *C. tropicalis*, *C. glabrata*, *C. krusei*, and *Candida guilliermondii*. We aimed to identify the etiologic agent as accurately and quickly as possible. The association of VVC/RVVC and some risk factors were also studied. Finally, we also characterized polymorphisms in genes related to azole resistance.

## 2. Patients and Methods

### 2.1. Vaginal Samples

A total of 129 vaginal samples were obtained from 115 women (aged >17 years) attending the outpatient clinic Bombero Etxaniz (Bilbao, Spain) from February 2013 to July 2015. Bombero Etxaniz has a midwifery unit and is the reference center for Sexually Transmitted Infections of the Biscay province. The research protocol was approved by the ethics committee of the Hospital of Basurto (the hospital of reference for this region) and the University of the Basque Country (UPV/EHU) (reference CEIAB/180//2014/MORAGUES TOSANTOS), and all subjects gave their informed consent to participate. Two samples were obtained from the lateral vaginal wall with a sterile cotton-tip swab. They were stored at 4 °C and processed no later than 48 h afterwards; one was sent to the reference hospital, and we processed the other. A case of VVC was defined as a patient with vulvar itching, vaginal discharge, and a positive culture.

### 2.2. Culture and Fungal Identification

Swab content was resuspended in 400 µL of sterile Phosphate-Buffered Saline (PBS) and divided into two aliquots. One of the aliquots was cultured on chromogenic agar (Condalab, Madrid, Spain) for 48 h at 37 °C, and the other one was stored at −20 °C for later multiplex qPCR identification.

#### Isolate Identification through PCR or ID32C

Green colonies grown on chromogenic medium were further identified using PCR following the protocol described by Romeo and Criseo (2008) to allow for discrimination between *C. albicans*, *Candida dubliniensis*, and *Candida africana* [20]. In brief, two colonies from fresh culture were added to 200 µL of PBS with glass beads; they were then vortexed and added directly to the PCR mixture, which included 10 µL of DNA Polymerase Biomix buffer (Bioline, London, UK) and 0.8 µL of each Cr primer (25 µM). Not-green colonies on chromogenic medium were identified using the ID32C system (bioMérieux, Marcy l’Etoile, France). Isolates not identified by any of the above-described methods were derived to rDNA amplification [21], and sequencing was performed at the SGIker for identification.

### 2.3. Molecular Identification of Fungal Species through Multiplex qPCR

#### 2.3.1. Automated DNA Extraction

Vulvovaginal samples were thawed, and then 90 mg of glass beads was added and vortexed for 5 min. Supernatants were transferred to a new collecting tube and 360 µL of MagNA Pure Bacteria Lysis Buffer (Roche, Mannheim, Germany) and 40 µL of Proteinase K (20 mg/mL) (Roche, Mannheim, Germany) were added. After incubating for 1 h at 65 °C and 10 min al 95 °C, automated DNA extraction was performed using 400 µL of the supernatant in the MagNA Pure Compact nucleic acid extraction instrument (Roche, Mannheim, Germany) using the protocol DNA_Bacteria_V3_2 and the extraction kit MagNA Pure Compact Nucleic Acid Isolation Kit I (Roche, Mannheim, Germany). The extracted DNA was eluted in 50 µL of buffer.

#### 2.3.2. Primer and Taqman Probe Design for qPCR Assays

Sequences of the 18S-ITS1-5.8S region were aligned using the MUSCLE software (version muscle3.8.31) [22]. All the sequences were obtained from the NCBI Entrez Nucleotide Database (https://www.ncbi.nlm.nih.gov/nucleotide/, accessed on 1 June 2010). Homologous areas of the 18S-ITS1-5.8S region were used to design Diamol-F and Diamol-R primers with the Primer Express software (version 3.0) (Applied Biosystems, Foster City, CA, USA). Furthermore, 12 specific Taqman probes that were designed were hybridized into the ITS1 region. Six of them were labeled with 5′-FAM, four with 5′-HEX, and two with 5′-JOE; all of them were labeled with 3′-BHQ1. On the other hand, an internal control (IC) of amplification and a probe (ICP) hybridizing this IC were designed to detect the inhibition of qPCR. The 5′ end of the IC included the same sequence as the primer Diamol-F, and the 3′ end included the reverse-complementary sequence of the primer Diamol-R. The ICP sequence was located four bases downstream of the Diamol-F primer sequence; the rest was a chimeric sequence designed not to hybridize with any of the target species, neither with cohabitating species of the vulvovaginal niches nor with human DNA. ICP was labeled with 5′-ROX and 3′-BHQ2. Table 1 lists the sequences of the primers and probes designed.

#### 2.3.3. Multiplex qPCR

All PCR protocols followed the recommendations of Khot and Fredricks (2009) to avoid both DNA contamination and PCR inhibition [23]. In brief, DNA extraction, PCR reaction mixes, amplification, and PCR product analysis were performed in different rooms. In addition, PCR reaction mixtures were performed in a laminar-flow biosafety cabinet, whose surfaces were cleaned with Termi-DNA-tor (Biotools, Madrid, Spain).

Before proceeding with multiplex assays, singleplex assays were carried out in a 7300 Real-Time Cycler (Applied Biosystem, Foster City, CA, USA). Amplification mixtures with a final volume of 25 µL contained 10 µL of Premix Ex TaqTM (Takara, Tokyo, Japan), 0.65 µM of each primer, 0.3 µM of each probe, and 1 pmol of the IC. Then, 20 nanograms of DNA extracted from microorganism cultures or 2 µL of DNA extracted from vaginal samples was added as template. To determine the analytical sensitivity and reproducibility of the qPCR, decreasing amounts of DNA, from 20 ng to 20 fg, extracted from cultures of 6 fungal species were tested. All samples were assayed in triplicate and on 3 different days. Afterwards, probes were combined in pairs, considering the fluorophore label.

### 2.4. Antifungal Susceptibility Testing

Fungal isolates were tested for in vitro susceptibility to fluconazole and clotrimazole (Sigma Aldrich-Merck, Madrid, Spain). Fluconazole testing was performed as described in the document M27, 4th ed., from the Clinical Laboratory Standards Institute (CLSI) [24], and result interpretations were performed according to the M27M44S document [25]. On the other hand, since there is no standardized procedure or minimal inhibitory concentration (MIC) limit ranges for microdilution test for clotrimazole, we followed the protocol proposed by Pelletier et al. (2000), which is based on the CLSI M27-A3 document with some modifications [26].

The isolates exhibiting reduced susceptibility to fluconazole (MIC ≥ 4 µg/mL) and/or clotrimazole (MIC ≥ 0.25 µg/mL) were also tested with the Sensititre YeastOne10^®^ (Trek Diagnostic System, East Grinstead, UK) colorimetric method in order to assess their susceptibility to other azoles.

### 2.5. Gene Expression Analysis through RT-qPCR

The expression levels of *CDR1*, *CDR2,* and *MDR1* genes were measured in three resistant (Be-113, Be-114, and Be-129), and three susceptible (SC5314, Be-47, and Be-90) yeast strains; for normalization purposes, the expression of two reference genes, *ACT1* (actin) and *PMA1* (H+-ATPase), was also measured [16,27] (Appendix A, upper panel). Total RNA was extracted from mid-log cultures in YPD (1% yeast extract, 2% peptone, 2% glucose) using the MasterPure™ Yeast RNA Purification Kit (Epicentre, Verona, WI, USA) following the manufacturer’s instructions with some modifications; proteinase K treatment was carried out at 50 °C for 20 min and DNase treatment was extended up to 30 min. Every isolate was extracted from three independent cultures and quantified with NanoDrop 1000 (ThermoFisher Scientific, Waltham, MA, USA). After checking the quality and integrity of the RNA, the cDNA was synthesized with PrimeScript™ RT reagent Kit (Perfect Real Time; Takara, Tokyo, Japan) following the manufacturer’s instructions, and negative controls (RT-minus) of the reverse-transcription were performed. RNA and cDNA samples were stored at −80 °C until use. RT-qPCR was performed in an Applied Biosystems 7300 Real-Time PCR System with SYBR^®^ Premix ExTaq™ (Tli RNaseH Plus; Takara, Tokyo, Japan). Technical triplicates of each sample were performed, and for each gene, a non-template control was included. The stability of the reference genes used for normalization was assessed with the web-based RefFinder tool (https://www.heartcure.com.au/reffinder/, accessed on 13 April 2022). The fold change was calculated with the 2^−ΔΔCt^ method [28]. RNA transcript levels of the resistant clinical isolates were compared to the average expression of the susceptible ones, which was set to 1, and each gene was considered to be overexpressed when the fold change in the expression was ≥2.

### 2.6. Sequencing Analysis of ERG11, TAC1, UPC2, MRR1, and MRR2 Genes

The *C. albicans* isolates that showed reduced or dose-dependent azole susceptibility were selected for the sequencing of resistance-related genes. DNA was extracted using the DNeasy^®^ Plant Mini Kit (Qiagen, Hilden, Germany), following the manufacturer’s instructions with modifications. In brief, 4-5 yeast colonies grown in Sabouraud agar O/N at 37 °C were suspended in 400 µL of sterile PBS, and 90 mg of sterile glass beads (Sigma Aldrich-Merck, Madrid, Spain) was added and vortexed for 5 min. The lysate was transferred to another tube, and 300 µL of Tris buffer (50 mM Tris-HCl pH8, 25 mM Na-EDTA, and 75 mM NaCl) and 15 U of lyticase were added. After an incubation at 30 °C for 20 min, 4 µL of RNAse (Qiagen, Hilden, Germany) was added and incubated at 37 °C for 20 min. Then, 100 µL of 10% SDS and 40 µL of Proteinase K (Roche, Mannheim, Germany) were added and incubated at 55 °C for 20 min. At this point, the manufacturer’s kit protocol was followed from the fourth step. The DNA sample was quantified in NanoDrop 1000 (ThermoFisher Scientific, Waltham, MA, USA) and stored at −20 °C until use.

The primers used for sequencing the *TAC1*, *ERG11*, *UPC2*, *MRR1,* and *MRR2* genes were designed to amplify the regions where mutations related to resistance to azoles had been previously identified in the literature; they are listed in Appendix A (lower panel) [29,30]. The sequences of the amplicons were provided by the Sequencing and Genotyping Service SGIker of the UPV/EHU. The analysis was conducted with Chromas 2.5.1 software (Technelysium, Queensland, Australia) to identify heterozygous mutations and, in order to identify the homozygous ones, BioEdit Sequence Alignment Editor 7.2.5 (Raleigh, NC, USA) was used to align the sequences obtained with reference sequences for *TAC1* (GeneBank accession no. DQ393587), for *UPC2* (GeneBank accession no. EU583451), for *ERG11* (GeneBank accession no. X13296), and for *MRR1* (GeneBank accession no. XM_711520) from Morio and collaborators (2013) [19], while for *MRR2*, the sequence with the GeneBank accession no. XM_705846 was used.

### 2.7. Statistical Analysis

We used the 2 × 2 contingency table to calculate sensitivity, specificity, and predictive values of the multiplex qPCR, and the X^2^ test for the association between the risk factors and VVC. *p* values ≤ 0.05 were considered statistically significant.

## 3. Results

Of the 129 isolates from the 115 patients enrolled in this study (4 isolates were obtained from the same patient, and from 11 other patients; 2 isolates from each were achieved), VVC was confirmed in 86 (66.67%) cases and, among them, 65 (75.58%) were classified as RVVC. The median age of the patients with VVC/RVVC was 30 years (range of 17 to 51 years), slightly lower than the total population analyzed, 32 years (range 17 to 57 years). Among the positive patients, 46 (53.49%) were pregnant, 17 (19.77%) had received antibiotic treatment in the last month, 13 (15.12%) had a history of allergic disorders, including allergic rhinitis, asthma, or sinusitis, 11 (12.8%) were taking oral contraceptives, 3 (3.49%) male partners of the patients with VVC had symptoms of *Candida* infection, and 1 (1.16%) was diabetic. Pregnancy was the only risk factor statistically associated to VVC/RVVC.

A total of 86 isolates were obtained from the 129 vaginal specimens, including 81 (94.2%) *C. albicans*, 2 (2.32%) *C. parapsilosis*, 2 (2.32%) *C. glabrata*, and 1 (1.16%) *C. tropicalis*. Only one specimen registered a mixed infection with *C. albicans* and *C. glabrata*.

### 3.1. Yeast Identification Using Multiplex qPCR

#### 3.1.1. Collection of Microorganisms

The Diamol primers pair and the combinations of Calb-Cpar3, Cgla-Cgui4, and Ctro2-Ckru2 probes allowed the specific detection of *C. albicans*–*C. parapsilosis*, *C. glabrata*–*C. guilliermondii*, and *C. tropicalis*–*C. krusei*, respectively (Table 2). Almost all the other species in our collection analyzed were negative for all the probes mentioned above, with the exception of the Cgla probe, which hybridized with *Saccharomyces cerevisiae* DNA. We decided to remove the IC, excluding the possibility of detecting qPCR inhibition, since probe-ICP had to be labeled with ROX; this impeded the detection of any signal when three probe pairs were combined.

On the other hand, Calb, Cpar3, and Cgla probes were tested in order to evaluate their specificity within the *C. albicans*, *C. parapsilosis*, and *C. glabrata* complexes, respectively (Appendix A). Cpar3 did not recognize *C. metapsilosis* or *C. orthopsilosis* cryptic species, while Calb and Cgla probes hybridized with all species of their respective complexes. New Calb2 and Cgla2 probes were designed; Calb2 did not hybridize with *C. dubliniensis*, while Cgla2 did not improve its specificity with respect to the Cgla probe.

#### 3.1.2. Analytical Sensitivity or Limit of Detection (LOD) of the Multiplex qPCR

Three independent standard curves ranging from 20 fg to 20 ng of DNA for *C. albicans*, *C. glabrata*, *C. parapsilosis*, *C. guilliermondii*, *C. tropicalis*, and *C. krusei* showed LOD of 20 fg for all the probes (Figure 1).

#### 3.1.3. Validation of the Multiplex qPCR with Vaginal Swab Samples/Analytical Sensitivity and Specificity of Multiplex qPCR Compared with Conventional Culture

The multiplex qPCR developed was applied to 129 human vulvovaginal samples. Along with these samples, DNA extraction controls were performed, and multiplex qPCR was applied to these controls in order to monitor fungal/DNA contamination and detect false positives.

Of the 129 vaginal swabs, 123 (96.1%) were PCR-positive for *C. albicans*, 3 (2.3%) for *C. glabrata*, 2 (1.6%) for *C parapsilosis* and *C. guilliermondii*, 3 (2.3%) for *C. tropicalis*, and 4 (3.1%) *C. krusei*. Given such a high rate of positivity, probably because of the non-sterile origin of the samples, a Ct cut-off value of 30 cycles was established, so that samples with Ct values above 30 were considered negative. Notably, PCR contamination could be excluded, as evidenced by the lack of amplification in non-template controls (NTC) for the six probes used. The majority of the species identified in the vaginal swabs through fungal culture and multiplex qPCR were *C. albicans*; therefore, we only determined the analytical sensitivity of the qPCR for the Calb probe. For that purpose, we built a contingency table (Appendix A); the analytical sensitivity of the qPCR for Calb probe using fungal culture as the gold standard was 91.35%, the analytical specificity was 89.6%, the positive predictive value was 93.67%, and the negative predictive value 86%.

We were able to detect the two *C. glabrata* culture-positive samples with the Cgla probe, but we obtained a discordant negative-culture, positive-PCR result in one sample. The Ctro2 probe detected the only culture-positive sample for *C. tropicalis*, but we were not able to detect the two *C. parapsilosis* culture-positive samples with the Cpar3 probe. Finally, the Cgui4 and Ckru2 probes did not hybridize with any sample, so no false positives were obtained for these probes.

### 3.2. Antifungal Susceptibility

A total of 78 isolates were tested for susceptibility to fluconazole and clotrimazole (8 isolates lost viability), MIC ranges, MIC50s, and MIC90s of *C. albicans* isolates are summarized in Appendix A. Most isolates were susceptible to both azoles. Three *C. albicans* isolates (Be-113, Be-114, and Be-129), showing high MICs of fluconazole (MIC ≥ 4 µg/mL) or clotrimazole (MIC ≥ 0.25 µg/mL), were also tested with caspofungin, micafungin, anidulafungin, amphotericin B, 5-flucytosine, voriconazole, posaconazole, and itraconazole using the Sensititre YeastOne method (Appendix A). Be-113 was resistant to posaconazole and itraconazole, and susceptibly dose-dependent to fluconazole, voriconazole and clotrimazole; Be-114 was susceptibly dose-dependent to fluconazole and clotrimazole. Be-113 and Be-114 showed an MIC of 0.25 µg/mL for clotrimazole, a notably higher value than the rest of the vulvovaginal isolates; therefore, they were considered susceptibly dose-dependent. Finally, Be-129 was resistant to fluconazole and voriconazole, and susceptibly dose-dependent to itraconazole. Related to NAC strains, the two *C. glabrata* isolates showed dose-dependent susceptibility to fluconazole and clotrimazole, while *C. parapsilosis* and *C. tropicalis* strains were susceptible to both azoles.

### 3.3. Expression of CDR1, CDR2, and MDR1 Genes of Azole-Resistant Isolates

We measured the expression levels of these genes, prior to sequencing, in order to identify which isolates could harbor GOF mutations in Tac1, Mrr1, and Mrr2 transcription factors. Be-113 overexpressed *CDR1* and *CDR2* (expression level > x2), while Be-114 reached an overexpression close to 2, and Be-129 did not overexpress any of the genes analyzed.

### 3.4. Amino Acid Substitutions in Erg11, Tac1, Upc2, Mrr1, and Mrr2

Be-113 and Be-129 isolates harbored mutations leading to amino acid substitutions in Erg11 (A114S, Y132H, Y257H, and G450E) and Upc2 (G648S, in heterozigosis) that have been previously described and related to azole resistance (Table 3) [16,19,31,32,33,34]. Moreover, Be-113 displayed a homozygous genetic alteration in Tac1, S758F, which is described for the first time in this study. Finally, the Be-114 isolate did not harbor any of the known resistance-related mutations that could be related to reduced azole susceptibility, but it had a new mutation, A311V.

## 4. Discussion

We developed a multiplex-qPCR assay to identify several *Candida* species simultaneously from vaginal swabs. The assay sensitivity and specificity for the *C. albicans* probe (Calb) and cut-off Ct value of 30 were 91.35% and 89.6%, respectively. Our results improve those obtained by Giraldo and colleagues (S = 29.4%) [35], Weissenbacher and colleagues (S = 42%) [36], and Mårdh and colleagues (E = 50%) [37] and are similar to the excellent results of Cartwright and colleagues [38] or, more recently, of Tardif and Schlaberg and Gaydos and colleagues [1,39]. Non-*albicans Candida* isolates were scarce, so we were unable to validate the qPCR assay for the other probes, but the Cgla and Ctro2 probes served to detect two infections caused by *C. glabrata* and one by *C. tropicalis*, respectively. The Cpar3 probe could not detect two samples of *C. parapsilosis*, probably because of the low fungal load of these samples (1/2 CFU). The Cgui4 and Ckru2 probes, designed to detect *C. guilliermondii* and *C. krusei*, did not cross-react with any of the clinical samples, highlighting the specificity of these two probes.

The differentiation of *Candida*/fungal species is clinically relevant, not only for epidemiological reasons but also for differences in virulence and antifungal susceptibility among species [40]. Accurate diagnosis is, therefore, of utmost importance to provide optimal treatment. We were able to detect cryptic species of the *C. albicans* and *C. parapsilosis* complexes since the Calb2 and Cpar3 probes only hybridized with their respective *C. albicans* and *C. parapsilosis* strains from our collection. We could not distinguish *C. albicans* and *C. africana* using the Calb probe, but both species are highly similar and several studies have also failed to achieve this milestone [40]. Furthermore, some authors do not even consider them different species [41]. Due to differences in virulence and antifungal susceptibility patterns among species within the same complex, the differentiation of *Candida* cryptic species has also become clinically relevant [40]. Finally, regarding the specificity of the multiplex qPCR assay, none of our probes hybridized with any of the tested microorganisms that can be present in human microbiota, not even with human DNA. An exception came from the Cgla probe, which cross-reacted with DNA from *S. cerevisiae*, a species much closer to *C. glabrata* than other species of *Candida* [42]. This mismatch implies that this probe must be optimized, and designing a new probe with locked nucleic acid (LNA) technology could be a good option.

Compared with culture, the gold standard for VVC [11], this multiplex qPCR assay might imply an improvement in diagnosing VVC, especially including accuracy, reliability, and the time needed to finish the tests. In this sense, our assay provided an answer for clinical swabs in 5 h compared to the 24–96 h required for culture detection [6]. Given this delay, many women do not seek medical diagnosis, resulting in underdiagnoses of VVC, making it impossible to perform epidemiologic studies of this infection [5]. In this study, 15/115 (13.04%) of the patients suspecting fungal infection that attended the outpatient clinic were finally diagnosed with *Gardnerella vaginalis*, a bacteria responsible for bacterial vaginosis. The later and VVC are common infections that are frequently misdiagnosed [43]. Additionally, the OTC availability of antifungal agents is convenient for rapid symptom relief, but it has also worsened the misdiagnosis of candidiasis. Furthermore, azole overuse and overexpose might be associated with a high resistance rate in *C. albicans* isolates, and several authors have reported a trend towards less susceptible vulvovaginal *Candida* spp. isolates [44,45,46]. Regarding antifungal susceptibility, we found three (3.48%) susceptible dose-dependent or azole-resistant *C. albicans* isolates. All of them were isolated from patients suffering from RVVC for a long time. Two of these isolates harbored mutations related to azole resistance: A114S and Y257H in Erg11p for the Be-113 isolate and Y132H and G450E in Erg11p and G648S in Upc2p for Be-129 [16,19,31,32,33,34]. The aforementioned mutations could explain the high MIC values of the Be-113 and Be-129 isolates and the treatment failure for these patients. On the contrary, the Be-114 isolate did not harbor any of the already known resistance-related mutations; first classified as clotrimazole SDD, it was reinterpreted with new CLSI cut-off values [25] and categorized as an azole-susceptible isolate. Nevertheless, since it was obtained from a patient who did not respond to azole treatment, we would expect Be-114 to be a resistant strain. Bearing in mind that the CLSI M27-A4 antifungal drug susceptibility test (AST) does not consider vulvovaginal pH and drug pharmacokinetics in this anatomical niche, Sobel and Akins compared MIC values for 125 *C. albicans* vaginal isolates at pH 7.0 and pH 4.5 and observed that the lower the pH, the higher the MIC values, as a result of the reduction in drug activity. Consequently, they proposed that AST for vulvovaginal isolates should be performed at pH 4.5 [10]. This could explain why the Be-114 isolate was reported by the lab as sensible, although it behaved as a clinically resistant isolate in situ.

To summarize, molecular characterization of resistance-related polymorphisms is plausible for the rapid diagnosis of resistance in *C. albicans* isolates. We used Sanger sequencing to identify mutations associated with resistance, which is laborious and does not permit the analysis of complete genes. However, probe-based real-time PCR assays are easier to perform, and when next-generation sequencing (NGS) platforms become cheaper, such broad analyses could replace or encompass culture-based diagnosis [47]. Here, we described two new mutations, S758F in Tac1p and A311V in Mrr2p, in two isolates with reduced susceptibility and/or treatment failure, whose putative involvement in fluconazole resistance must be empirically tested. To this end, Clustered Regularly Interspaced Short Palindromic Repeats-Cas9 (CRISPR-Cas9) seems a promising technology. In addition, vulvovaginal *C. albicans* isolates may be considered a model to detect new mutations and/or mechanisms related to azole resistance or tolerance, since they are faced to repeated short courses of low antifungal concentrations, favoring their adaptation to azole pressure [48].

This study has a few noteworthy limitations. Due to the project’s conclusion, we could not reach the minimum number of samples stated to the ethics committee. Furthermore, although the Bombero Etxaniz outpatient clinic is a reference center for Biscay, the geographical area should have been expanded to obtain more representative data. Finally, concerning the molecular analysis of azole resistance, we analyzed gene regions linked to decreased susceptibility reported in the literature instead of complete genes, which could imply that we missed more GOF mutations.

## 5. Conclusions

In conclusion, our multiplex-qPCR assay provides a rapid method to accurately diagnose vulvovaginal infections caused by *C. albicans*, a condition usually underdiagnosed. Furthermore, the existence of resistance-related mutations of *C. albicans* isolates from this anatomic origin emphasizes the need for precise resistance diagnosis, and molecular characterization holds promise for rapid reporting.

## Figures and Tables

**Figure 1 jof-09-01145-f001:**
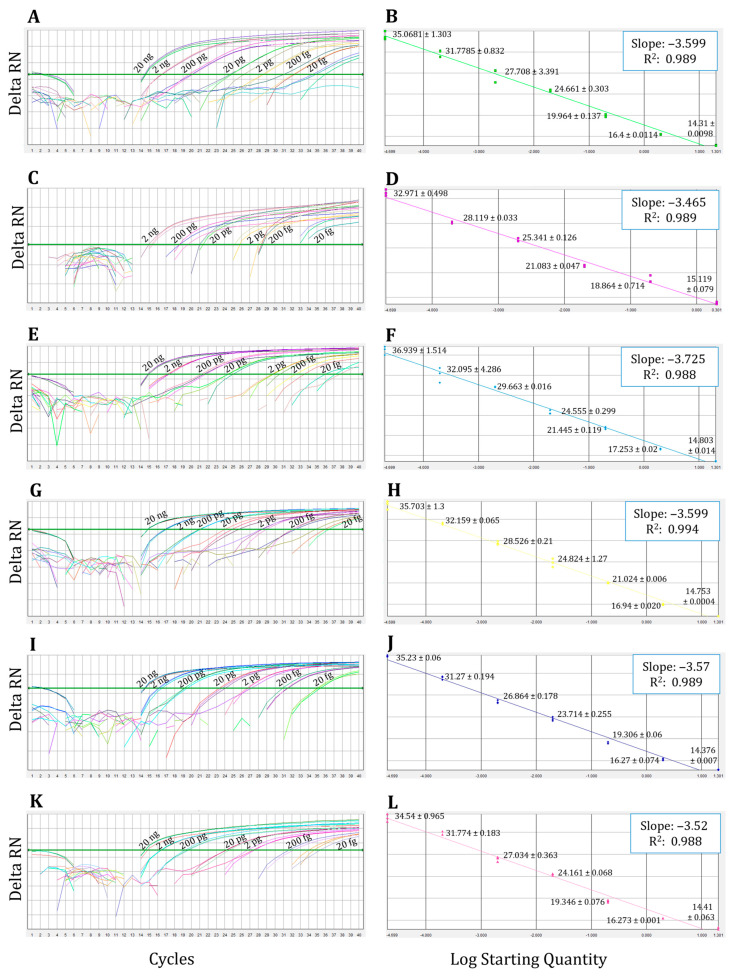
Results of the representative standard curves with de singleplex qPCR assay: Logarithmic representation of fluorescence detected with (**A**) Calb, (**C**) Cgla, (**E**) Cpar3, (**G**) Cgui4, (**I**) Ctro2, and (**K**) Ckru2 probes with the *C. albicans*, *C. glabrata*, *C. parapsilosis*, *C. guilliermondii*, *C. tropicalis*, and *C. krusei* (respectively) DNA serial dilutions from 20 fg to 20 ng. (**B**,**D**,**F**,**H**,**J**,**L**) Plot of the Ct values versus the logarithmic of the genomic DNA starting concentration. Numbers next to replicates indicate the detection threshold cycles’ mean values obtained for each DNA concentration ± SD.

**Table 1 jof-09-01145-t001:** Sequences of primers and probes used in the multiplex qPCR assays.

Primers and Probes	Species	Gene	Sequences (5′→3′)
Primers			
Diamol-F	-	18S	TAGGTGAACCTGCGGAAGGA
Diamol-R	-	5.8S	TCGCTGCGTTCTTCATCGAT
Probes			
Calb	*C. albicans*	ITS1	FAM-CGGTGGGCCCAGCCTGCC-BHQ1
Calb2	*C. albicans*		FAM-ATCAA[C]TTGTCACA[C][C]AGA-ZNA4-BHQ1
Cpar2	*C. parapsilosis*		JOE-AGGCC[C]CATA[T]AGAAGG[C]CTA-BHQ1
Cpar3	*C. parapsilosis*		HEX-TGGCAGGCCCCATATAGAAGGCCTAC-BHQ1
Cgla	*C. glabrata*		FAM-ATTTCTCCTGCCTGCGCTTAAGTGCG-BHQ1
Cgla2	*C. glabrata*		FAM-TTAAGTGCGCGG[T][T]GGTGG-ZNA4-BHQ1
Cgui3	*C. guilliermondii*		JOE-AA[C]CTA[T]CT[C]TA[G]GC[C]AAA-BHQ1
Cgui4	*C. guilliermondii*		HEX-CAGCGTTTAACTGCGCGGCGA-BHQ1
Ctro	*C. tropicalis*		HEX-CGGTAGGATTGCTCCCGCCA-BHQ1
Ctro2	*C. tropicalis*		HEX-CGGTAGGATTGCTCCCGCCACC-BHQ1
Ckru	*C. krusei*		FAM-TTTAGGTGTTGTTGTTTTCGTTCCGCTC-BHQ1
Ckru2	*C. krusei*		FAM-CTACACTGCGTGAGCGGAACGAAAAC-BHQ1
Control			
Probe-ICP	-	-	ROX-AACGTGCGACGTTCCGAGCA-BHQ2
IC	-	-	*TAGGTGAACCTGCGGAAGGA*TCGA**AACGTGCGACGCTTCCGAGCA**TGATCACTATGTCCTAATCCCATATATTATTCACTGTGTACTAGCCCTTCTTGGTTCTCGC*ATCGATGAAGAACGCAGCGA*

Probes were marked with different fluorescent reporters attached at the 5′ end (FAM, JOE, HEX, and ROX) and Black Hole Quencher (BHQ) at the 3′ end. These were supplied by Metabion International AG, (Munich, Germany), Integrated DNA Technologies (IDT, Madrid, Spain), and/or Sigma-Aldrich-Merck (Madrid, Spain). The internal control (IC) 5′ end includes the same sequence (italic letters) of the primer Diamol-F, and the 3′ end includes the reverse-complementary sequence of the primer Diamol-R; the Probe-ICP sequence is highlighted in bold. Locked nucleic acids are shown in square brackets [ ].

**Table 2 jof-09-01145-t002:** Results obtained through multiplex qPCR with some species from our microorganism collection including Ct mean values ± SD.

		Probes (Ct ± SD) ^b^					
Species	Strain ^a^	Calb	Cgla	Cpar3	Cgui4	Ctro2	Ckru2
*C. albicans*	NCPF 3153	14.26 ± 0.056	-	-	-	-	-
*C. glabrata*	NCPF 3203	-	13.34 ± 0.056	-	-	-	-
*C. parapsilosis*	NCPF 3104	-	-	14.4 ± 0.106	-	-	-
*C. guilliermondii*	NCPF 3099	-	-	-	14.33 ± 0.891	-	-
*C. tropicalis*	NCPF 3111	-	-	-	-	14.31 ± 0.993	-
*C. krusei*	ATCC 6258	-	-	-	-	-	13.64 ± 0.007
Other yeasts							
*Saccharomyces cerevisiae*	CECT 1678	-	18.37	-	-	-	-
*Magnusiomyces capitata*	IHEM 5666	-	-	-	-	-	-
*Yarrowia lipolytica*	UPV 12-097	-	-	-	-	-	-
*Rhodotorula mucilaginosa*	CECT 11016	-	-	-	-	-	-
Filamentous fungi							
*Aspergillus fumigatus*	Af-293	-	-	-	-	-	-
*Lomentospora prolificans*	ATCC 64913	-	-	-	-	-	-
*Cryptococcus neoformans*	ATCC 90113	-	-	-	-	-	-
Bacteria							
*Staphylococcus aureus*	CECT 435	-	-	-	-	-	-
*Streptococcus pyogenes*	CECT 985	-	-	-	-	-	-
*Streptococcus viridans*	CECT 804	-	-	-	-	-	-
*Streptococcus pneumoniae*	CECT 993	-	-	-	-	-	-
*Escherichia coli*	CECT 434	-	-	-	-	-	-
*Klebsiella pneumoniae*	CECT 144	-	-	-	-	-	-
*Pseudomonas aeruginosa*	CECT 108	-	-	-	-	-	-
*Proteus mirabilis*	CECT 4168	-	-	-	-	-	-
*Gardnerella vaginalis*	ATCC 14018	-	-	-	-	-	-
Human DNA							
Human Genomic DNA (Promega, Spain)	G304A *	-	-	-	-	-	-

^a^ ATCC = American Type Culture Collection; CECT = Colección Española de Cultivos Tipo; IHEM = Belgian Coordinated Collections of Microorganisms; UPV = Collection from the University of the Basque Country (UPV/EHU). ^b^ Mean Ct and SD values in three replicates on three different days. * Catalog number from Promega. (-) No detection.

**Table 3 jof-09-01145-t003:** Amino acid substitutions in the transcription factors Tac1p, Upc2p, Mrr1p, and Mrr2p and in Erg11p.

Isolate	Erg11p ^a^	Tac1p ^b^	Upc2p ^c^	Mrr1 ^d^	Mrr2 ^e^
Be-113	**A114S**; **Y257H**	N396S ^h^; S758F	-	V341E ^h^; L592F ^h^; E1020Q ^h^	-
Be-114	-	A337V ^h^; N396S; N772K; D776N; E829Q ^h^; S941P ^h^	-	V341E; E1020Q	A311V; A451A; V582L
Be-129	**Y132H**; **G450E**	N396S ^h^; N772K ^h^; D776N ^h^; E829Q ^h^; S935L ^h^; S941P ^h^	**G648S ^h^**	-	-

^a^ The nucleotide coding sequences spanning the following positions were analyzed: nucleotides 83-1554 for *ERG11*; ^b^ nucleotides 673-1383 and 2017-2940 for *TAC1*, ^c^ nucleotides 1806-2094 for *UPC2*, ^d^ nucleotides 908-2214 and 2266-3200 for *MRR1*, ^e^ and full-length for *MRR2*. Each aa involved in resistance as a GOF is highlighted in bold, new mutations are underlined. ^h^ heterozygous (mutation in a single allele).

## Data Availability

Data are contained within the article.

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
