# Peer review of "Molecular Identification of Fungal Species through Multiplex-qPCR to Determine Candidal Vulvovaginitis and Antifungal Susceptibility"

_jof, 2023, doi:10.3390/jof9121145_

Round 1

Reviewer 1 Report

Comments and Suggestions for Authors

General considerations

Introduction. Consider to add more specific information about the etiology of VVC, the role of microbiological diagnosis (microscopy, vaginal pH), the issue of colonization (especially in RVVC) and basic information about treatment (available antifungals, management of acute vs. recurrent VVC).

Conclusion. Missing in abstract. Due to the limited number of patients and vaginal samples (and high prevalence of C. albicans), caution is warranted against generalization; the main impact of the introduction of the multiplex qPCR methodology is on the diagnosis of C. albicans, more robust data are lacking for other species.

Corrections/ suggestions

Line 19: Add % of C. albicans.

Line 116/ 209/ 221: Better present yeast species by (clinically) best-known names (which are acceptable and valid - see de Hoog S. et al. 2023) i.e., Candida glabrata, C. guilliermondii, C. krusei.

Line 146: Provide the MIC value of fluconazole and clotrimazole for resistance (reason for retesting).

Line 221: Add footnotes a and b.

Line 234: Add full stop.

Line 239, 240, 241: Unify the number of decimal places.

Line 242: Replace the archaic “stablished” with “established”.

Line 260, 267: Species names should in italics.

Line 269: In parentheses, indicate the MIC interval for the resistance of fluconazole and clotrimazole.

Line 294: Footnotes (a and h) are missing.

Line 350: The explanation is not entirely clinically satisfactory, because if the choice of therapy is based on the results of antifungal susceptibility testing (and it usually is, especially, in the case of RVVC), there is a risk of false interpretation and failing therapy. One of the ways to avoid it is to extend the result reading to 48 hours.

Table S2:

Correct nivariensisa on nivariensis.

Correct bracarensisa on bracarensis.

Saprochaete capitata is now not considered a correct name, replace it for Magnusiomyces capitatus

(de Hoog S et al. 2023, J. Clin. Microbiol., doi 10.1128/jcm.00873-23)

Table S4:

The subject of interest (detection of yeasts) is missing from the title.

Specify chromogenic medium (supplier or manufacturer).

Table S5:

The subject of interest (Candida albicans) is missing from the title.

Correct “bol” on “bold” (last word in footnotes).

Comments on the Quality of English Language

No comments

Author Response

1st REVIEWER. SUMMARY:

Thank you very much for taking time to review our manuscript. Please find the detailed responses below, in red, and the corresponding revisions and corrections highlighted, in yellow, in the re-submitted files.

General considerations

Introduction. Consider to add more specific information about the etiology of VVC, the role of microbiological diagnosis (microscopy, vaginal pH), the issue of colonization (especially in RVVC) and basic information about treatment (available antifungals, management of acute vs. recurrent VVC). ANSWER: We appreciate the Reviewer’s comment, the required information about aetiology, antifungals and RVVC, and diagnosis have been added (in the case of this last item, only a brief description has been written; since we are not medical practitioner, we hope the reviewer will understand our caution about this topic) (lines 40-47, 55-59).

Conclusion. Missing in abstract. Due to the limited number of patients and vaginal samples (and high prevalence of C. albicans), caution is warranted against generalization; the main impact of the introduction of the multiplex qPCR methodology is on the diagnosis of C. albicans, more robust data are lacking for other species. ANSWER: We agree with the Reviewer. We added a conclusion sentence to the Abstract (lines 22-24) and modified it following the Reviewer’s suggestions (lines 413-415).

Corrections/ suggestions

Line 19: Add % of C. albicans. ANSWER: missing data has been added (line 19).

Line 116/ 209/ 221: Better present yeast species by (clinically) best-known names (which are acceptable and valid - see de Hoog S. et al. 2023) i.e., Candida glabrataC. guilliermondiiC. krusei. ANSWER: We agree with the Reviewer, and all fungal names have been changed according to the recommended reference.

Line 146: Provide the MIC value of fluconazole and clotrimazole for resistance (reason for retesting). ANSWER: We appreciate the Reviewers’ comment, and MIC values have been added (lines 165, 166).

Line 221: Add footnotes a and b. ANSWER: Missing footnotes have been added (lines 249-251).

Line 234: Add full stop. ANSWER: Full stop has been added.

Line 239, 240, 241: Unify the number of decimal places. ANSWER: they have been unified (lines 269-271).

Line 242: Replace the archaic “stablished” with “established”. ANSWER: It has been replaced (lines 272).

Line 260, 267: Species names should in italics. ANSWER: We are sorry for the mistake; names have been modified.

Line 269: In parentheses, indicate the MIC interval for the resistance of fluconazole and clotrimazole. ANSWER: We appreciate the Reviewers’ comment, and MIC values have been added (lines 297, 298).

Line 294: Footnotes (a and h) are missing. ANSWER: We apologize to the reviewer for this mistake; footnotes a, b, c, d, e, and h have been added (lines 325-328).

Line 350: The explanation is not entirely clinically satisfactory, because if the choice of therapy is based on the results of antifungal susceptibility testing (and it usually is, especially, in the case of RVVC), there is a risk of false interpretation and failing therapy. One of the ways to avoid it is to extend the result reading to 48 hours. ANSWER: We did not have the opportunity to know if susceptibility testing was done before starting antifungal treatment. The reference hospital for this outpatient clinic (Basurto University Hospital) and we made a diagnosis in parallel (this information has been added to section “2.1 Vaginal samples” (lines 81, 82 and 87, 88)). The hospital only shared with us the results of the culture. In addition, the investigators involved in this study only knew the patients from a survey and did not have access to their clinical history. Be-114 patient stated that she had been using clotrimazole for years and that it had stopped working for her; in fact, her degree of discomfort was such that she had participated in a clinical trial to test the efficacy of a vaccine to prevent VVC. He had been free of Candida for two months, after which he had returned to Bombero-Etxaniz's office. We regret we cannot provide the reviewer with more information on this matter.

On the other hand, the sensitivity test reading was performed following the CLSI recommendations for azole compounds, which indicate that only if the growth in the control is not sufficient at 24 hours the interpretation should be made at 48 hours. Clotrimazole is not included in this test, but the interpretation was performed similarly to fluconazole.

Table S2:

Correct nivariensisa on nivariensis. ANSWER: We agree with the Reviewer, and fungal name has been changed according to the recommended reference

Correct bracarensisa on bracarensis. ANSWER: We agree with the Reviewer, and fungal name has been changed according to the recommended reference

Saprochaete capitata is now not considered a correct name, replace it for Magnusiomyces capitatus ANSWER: We agree with the Reviewer, and fungal name has been changed according to the recommended reference

(de Hoog S et al. 2023, J. Clin. Microbiol., doi 10.1128/jcm.00873-23)

Table S4:

The subject of interest (detection of yeasts) is missing from the title. ANSWER: subject of interest has been added.

Specify chromogenic medium (supplier or manufacturer). ANSWER: it has been completed.

Table S5:

The subject of interest (Candida albicans) is missing from the title. ANSWER: subject of interest has been added.

Correct “bol” on “bold” (last word in footnotes). ANSWER: it has been changed.

Comments on the Quality of English Language

No comments

Submission Date

25 October 2023

Date of this review

03 Nov 2023 07:14:01

Thank you so much for your comments and interesting suggestions. Yours sincerely.

Dr. Inés Arrieta-Aguirre

Reviewer 2 Report

Comments and Suggestions for Authors

The identification of Candida species responsible for VVC is relevant as they present different virulence characteristics and variable susceptibility to antifungals, which directly implies the choice of appropriate treatment.

Rapid and accurate diagnosis is important to allow early treatment of the infection, preventing the development of resistance.

The identification of mutations occured in genes responsible for resistance to antifungals is very useful for understanding the resistance development and this work also identified mutations not previously reported.

Another positive factor of this work was the high number of samples analyzed and the identification of species not common in this infection site, which are rarely reported, also showing the impact on their treatment as they are low susceptible to conventional treatment.

Minor corrections:

1-      Legends of Table 2 and Table 3 are missing

2-      Lines 226 and 227: “New Calf2 and Cglab2 new...” the word “new” is repeated, please correct it.

Author Response

2ND REVIEWER

Thank you very much for taking time to review our manuscript. Please find the detailed responses below, in red, and the corresponding revisions and corrections highlighted, in yellow, in the re-submitted files.

Comments and Suggestions for Authors

The identification of Candida species responsible for VVC is relevant as they present different virulence characteristics and variable susceptibility to antifungals, which directly implies the choice of appropriate treatment.

Rapid and accurate diagnosis is important to allow early treatment of the infection, preventing the development of resistance.

The identification of mutations occured in genes responsible for resistance to antifungals is very useful for understanding the resistance development and this work also identified mutations not previously reported.

Another positive factor of this work was the high number of samples analyzed and the identification of species not common in this infection site, which are rarely reported, also showing the impact on their treatment as they are low susceptible to conventional treatment.

Minor corrections:

1-      Legends of Table 2 and Table 3 are missing ANSWER: We apologize for this mistake; we have added missing captions (lines 249--251 and 325-328).

2-      Lines 226 and 227: “New Calf2 and Cglab2 new...” the word “new” is repeated, please correct it. ANSWER: we appreciate Reviewer’s observation; we have eliminated the extra word.

Submission Date

25 October 2023

Date of this review

07 Nov 2023 22:37:58

Thank you so much for your kind comments and interesting suggestions. Yours sincerely.

Dr. Inés Arrieta-Aguirre

Reviewer 3 Report

Comments and Suggestions for Authors

This manuscript, authored by Inés Arrieta-Aguirre and colleagues, presents a study on the molecular identification of fungal species in Candidal Vulvovaginitis specimens using a multiplex qPCR assay. The research also uncovered reduced azole susceptibility in a small percentage of isolates, pinpointing genetic mutations associated with drug resistance, including novel mutations in the TAC1 and MRR1 genes. It is well written and easy to read. Still, some minor aspects need to be fixed:

INTRODUCTION
- "women" is sociology > "female" is biology
- this fact translates into disease > invasive or non invasive?
- it is widely recognized that misdiagnosis of vaginal complaints is extremely common > why?
- ticularly Nakaseomyces glabrata (formerly known as Candida glabrata), Candida tropicalis or Pichia kudriavcevii (formerly known as Candida krusei) > suggest to cite the manuscript with the taxonomy changes

METHODS
- outpatient clinic Bombero-Etxaniz (Bilbao, Spain) > why this centre?
- was approved by the ethics committee > could authors provide approval codes?
- Condalab, Spain > Location? (this applies to the other companies mentioned in the manuscript too)
- C. albicans, Candida dubliniensis and Candida africana [16]. Briefly, two colonies > keep consistency in pathogen abbreviation

RESULTS
- 129 isolates and 115 patients > how many patients had double/triple isolates?

DISCUSSION
- Please, include a limitation section

Author Response

3RD REVIEWER

Thank you very much for taking time to review our manuscript. Please find the detailed responses below, in red, and the corresponding revisions and corrections highlighted, in yellow, in the re-submitted files.

Comments and Suggestions for Authors

This manuscript, authored by Inés Arrieta-Aguirre and colleagues, presents a study on the molecular identification of fungal species in Candidal Vulvovaginitis specimens using a multiplex qPCR assay. The research also uncovered reduced azole susceptibility in a small percentage of isolates, pinpointing genetic mutations associated with drug resistance, including novel mutations in the TAC1 and MRR1 genes. It is well written and easy to read. Still, some minor aspects need to be fixed:

INTRODUCTION
- "women" is sociology > "female" is biology. ANSWER: We acknowledged the Reviewer’s comment, but according to the Oxford Advanced Learner’s Dictionary a WOMAN is “an adult female human being”. Moreover, the term woman is widely employed in scientific publications; in fact, we have used women based on published articles, so since these are quotations (DOIs are included in this response), we do not know if it is appropriate to change women to female. Therefore, we still defend it in our manuscript.

[2]: https://www.sciencedirect.com/science/article/pii/S0140673607609179

[3]: https://www.sciencedirect.com/science/article/pii/S1473309918301038

[4]:https://journals.lww.com/greenjournal/fulltext/2012/12000/fluconazole_resistant_candida_albicans.22.aspx

[5]: https://www.sciencedirect.com/science/article/pii/S0002937815007164

- this fact translates into disease > invasive or non invasive? ANSWER: In this case, we are referring to Non-invasive disease. Nevertheless, we have modified the sentence in order to make it easier to understand (lines 33-34).

- it is widely recognized that misdiagnosis of vaginal complaints is extremely common > why? ANSWER: As Sobel and colleagues declare, it seems that the reluctance of many clinicians to perform any test at all, together with the belief that a vaginal yeast infection can be only diagnosed by history, has made this condition to be misdiagnosed (https://doi.org/10.1007/s11908-015-0488-3). Not only that, several studies have shown the existence of this bias in the diagnosis of vulvovaginal candidiasis (https://doi.org/10.1186/1476-0711-5-4). It should not be forgotten that the delay in diagnosis (48-72h) and the availability of over-the-counter antifungals lead patients not to seek clinical advice (https://doi.org/10.1016/j.ajog.2015.06.067).

- ticularly Nakaseomyces glabrata (formerly known as Candida glabrata), Candida tropicalis or Pichia kudriavcevii (formerly known as Candida krusei) > suggest to cite the manuiisscript with the taxonomy changes ANSWER: Following the recommendations of another reviewer and the publication of de Hoog et al (2023), we have changed them to the clinically best-known names of yeasts species: C. glabrata, C. krusei and C. guilliermondii (https://doi.org/10.1128/jcm.00873-23).

METHODS
- outpatient clinic Bombero-Etxaniz (Bilbao, Spain) > why this centre? ANSWER: Acknowledging the reviewer’s concern about the selection of this center, we have included the following sentence in the manuscript: “this clinic has a midwifery unit and is the reference center for Sexually Transmitted Infections of the Biscay province” (lines 81-82)

- was approved by the ethics committee > could authors provide approval codes? ANSWER: Sorry, we forgot to include the reference of the Ethics Committee approved protocol: CEIAB/180/2014/MORAGUES TOSANTOS. The reference has been added to the manuscript (lines 84-85).

- Condalab, Spain > Location? (this applies to the other companies mentioned in the manuscript too) ANSWER: We have added the missing data of the companies, as suggested by the reviewer.

- C. albicans, Candida dubliniensis and Candida africana [16]. Briefly, two colonies > keep consistency in pathogen abbreviation. ANSWER: We wrote the full names of Candida dubliniensis and Candida africana because it was the first time they were mentioned in the text, as opposed to Candida albicans, which had been previously named.

RESULTS
- 129 isolates and 115 patients > how many patients had double/triple isolates? ANSWER: We regret for not providing the detailed information. We registered four Gardnerella vaginalis isolates in serial samples from the same patient with recurrent infections, but Candida spp. were never reported. In addition, from 11 other patients, we obtained two isolates from each. Information has been added to the main text (lines 223-225).

DISCUSSION
- Please, include a limitation section ANSWER: In agreement with the reviewer’s recommendation, we have included a limitation section at the end of the discussion (lines 404-410).

Thank you so much for your comments and interesting suggestions. Yours sincerely.

Dr. Inés Arrieta-Aguirre